# Data-Independent Acquisition (DIA)-Based Proteomics for the Identification of Biomarkers in Tissue Washings of Endometrial Cancer

**DOI:** 10.3390/ijms262311498

**Published:** 2025-11-27

**Authors:** Lorenzo Monasta, Valeria Capaci, Feras Kharrat, Milena Ciampechini, Nour Balasan, Andrea Conti, Valentina Golino, Pietro Campiglia, Michelangelo Aloisio, Danilo Licastro, Giovanni Di Lorenzo, Federico Romano, Giuseppe Ricci, Blendi Ura

**Affiliations:** 1Institute for Maternal and Child Health, IRCCS ‘Burlo Garofolo’, 65/1 Via dell’Istria, 34137 Trieste, Italy; lorenzo.monasta@burlo.trieste.it (L.M.); feras.kharrat@burlo.trieste.it (F.K.); milena.ciampechini@burlo.trieste.it (M.C.); nour.balasan@burlo.trieste.it (N.B.); andrea.conti@burlo.trieste.it (A.C.); federico.romano@burlo.trieste.it (F.R.); giuseppe.ricci@burlo.trieste.it (G.R.); blendi.ura@burlo.trieste.it (B.U.); 2Department of Pharmacy, University of Salerno, Via Giovanni Paolo II 132, 84084 Salerno, Italy; valentina.golino@unina.it (V.G.); pcampiglia@unisa.it (P.C.); 3National PhD Program in RNA Therapeutics and Gene Therapy, University of Naples Federico II, 80131 Napoli, Italy; 4Functional Gastrointestinal Disorders Research Group, National Institute of Gastroenterology-IRCCS “Saverio de Bellis”, 70013 Castellana Grotte, Italy; michelangelo.aloisio@irccsdebellis.it; 5AREA Science Park, 34149 Trieste, Italy; licastrod@gmail.com; 6Department of Medicine, Surgery and Health Sciences, University of Trieste, 34149 Trieste, Italy

**Keywords:** endometrial cancer, proteomics, mass spectrometry, tissue washings, biomarker

## Abstract

Endometrial cancers (ECs) are mainly adenocarcinomas arising from the uterine endometrium. In this work, we employed data-independent acquisition (DIA) mass spectrometry (MS)-based label-free quantification (LFQ-MS) proteomics to analyze the proteome of tissue washings collected from 25 control (CTRL) subjects, 25 patients with low-grade type 1 endometrial cancer (EC), and 24 patients with high-grade type 1 EC. Following quantification and statistical analysis, we identified 42 proteins able to discriminate CTRL from EC patients, and 151 proteins differentiating high-grade EC cases from low-grade EC cases. Notably, *PRRC2A* and *SYDE2* effectively distinguished both EC patients from controls and advanced EC cases from low-grade EC cases. Validation by Western blot analysis in an independent cohort comprising 19 CTRL patients, 19 patients with low-grade EC, and 19 patients with high-grade EC confirmed the upregulation of *PRRC2A* and *SYDE2*. These proteins are implicated in the translocation of *SLC2A4*, the regulation of *MECP2*, and extracellular matrix (ECM) proteoglycan pathways, all of which are associated with tumor growth. Our results demonstrate that DIA-based proteomic analysis of tissue washings enables the identification of potential biomarkers for endometrial cancer (EC). Moreover, this study highlights tissue washings as a promising biological fluid for biomarker discovery in EC.

## 1. Introduction

Endometrial cancer (EC) is the most common gynecological malignancy in developed countries, ranked as the seventh most prevalent female cancer worldwide [1]. The incidence of EC has been steadily rising, especially in high-income countries, largely due to aging populations and increasing obesity rates, representing a significant health concern. Globally, there are over 400,000 new cases and more than 90,000 deaths from EC annually. In Europe, approximately 130,051 women in 2022 were diagnosed with endometrial cancer, with 29,963 associated fatalities [2,3]. Data from the same year, 2022, showed that EC’s prevalence was highest in Central Europe, and mortality was highest in Central and Eastern Europe. However, with nearly 80–85% surviving five years after diagnosis, women with EC diagnosed in Europe are in a relatively good situation [4]. EC is typically classified into two main subtypes according to the histopathological characteristics [5]: type 1 (estrogen-dependent, low-grade) and type 2 (non-estrogen-dependent, high-grade). Type 1 accounts for approximately 80% of cases and generally has a better prognosis. In contrast, type 2 is more aggressive and often diagnosed at an advanced stage.

Despite advances in treatment, survival rates have not significantly improved for advanced-stage EC, highlighting the need for earlier detection and better biomarkers.

Significant efforts are underway to identify endometrial cancer (EC) biomarkers to improve early detection, diagnosis, patient stratification, prognosis, and treatment. Biomarkers offer a promising solution by enabling non-invasive or minimally invasive diagnosis through the analysis of blood, uterine fluid, or tissue washings [6].

In a previous study by our group based on analyzing the proteomic profile of the exosome proteins derived from serum samples, eight proteins (*APOA1*, *HBB*, *CA1*, *HBD*, *LPA*, *SAA4*, *PF4V1*, and *APOE*) were identified to have diagnostic potential for the EC diagnosis [7]. Aside from using serum samples, uterine aspirates were used to identify biomarkers of the EC, and 25 proteins were found to have a diagnostic potential for the EC [8].

A recent study showed that a five-biomarker panel of cervicovaginal fluid-derived proteins (*HPT*, *LG3BP*, *FGA*, *LY6D*, and *IGHM*) predicted endometrial cancer with an AUC of 0.95 (0.91–0.98), a sensitivity of 91% (83–98%), and a specificity of 86% (78–95%) [9].

By analyzing urine samples, Njoku and colleagues [10] developed a diagnostic model of a 10-marker panel combining *SPRR1B*, *CRNN*, *CALML3*, *TXN*, *FABP5*, *C1RL*, *MMP9*, *ECM1*, *S100A7*, and *CFI*, and predicted endometrial cancer with an AUC of 0.92 (0.96–0.97). Urine-based protein signatures showed good accuracy for the detection of early-stage cancers [AUC 0.92 (0.86–0.9)].

Recently, tissue wash fluid was considered to analyze the proteomic profile. In a study to predict biomarkers for colon cancer, Giust and colleagues [11] identified 19 differentially expressed proteins by analyzing the washing fluids of the colon cancer samples derived from surgical operations.

To date, the potential of tissue washings for predicting biomarkers of EC has been largely overlooked. Tissue washings can be collected through minimally invasive routine procedures and accurately reflect the local tumor environment, making them valuable for biomarker discovery [12].

Therefore, our current study might hold important potential in the field of EC biomarkers. This study aimed to characterize the tissue fluid proteome from the EC and healthy individuals using high-resolution mass spectrometry HRMS, to identify novel EC biomarkers.

## 2. Results

### 2.1. Proteomics Study from EC Tissue Washes

The protein composition of 74 tissue wash samples (25 CTRL, 25 EC type 1 low-grade, and 24 EC type 1 high-grade) was analyzed. For DIA file processing we used Spectronaut 19 and identified 4255 (Appendix A) proteins with q < 0.01 and FDR < 1%. After removing plasma proteins, keratins, and hornerin, 4112 cellular proteins were used for the subsequent analyses (Appendix A). To assess differences in sample composition across experimental conditions, principal component analysis (PCA) was performed (Figure 1). The PCA scores plot revealed distinct clustering among the groups, with the first two principal components explaining a total of 50.9% of the variance (PC1: 33.6%, PC2: 17.3%). Notably, the control group (red) was clearly separated from both the EC high (green) and EC low (blue) groups along PC1, indicating significant compositional differences. These differences were statistically confirmed by PERMANOVA (F-value: 1228.7; R-squared: 0.97192; *p*-value (based on 999 permutations): 0.001). To further enhance discrimination between groups, partial least squares discriminant analysis (PLS-DA) was applied (Figure 1). PLS-DA provided improved separation compared to PCA, with distinct clustering of the EC high, EC low, and CTRL samples.

From the 4112 cellular proteins initially identified, we applied the Mann–Whitney U test (*p* < 0.05) combined with a fold change threshold (≥1.5 or ≤0.6) to determine significantly dysregulated proteins. In the comparison between high EC and low EC, we identified 2426 significantly dysregulated proteins, with 1241 upregulated and 1183 downregulated. In the comparison between high EC and CTRL, 2940 proteins were significantly dysregulated, of which 2386 were upregulated and 554 were downregulated. Finally, in the comparison between low EC and CTRL, 2903 dysregulated proteins were identified, including 2338 upregulated and 554 downregulated (Figure 2). In total, we identified 3661 significantly dysregulated proteins in all the comparisons.

A heatmap (Figure 3) with clustering displays of the 3661 dysregulated proteins across the three comparisons, highlighting their correlation patterns.

### 2.2. Statistical Analysis

The 42 proteins that completely separated ECs from CTRLs are listed in Appendix A. For each variable we report the median, minimum, maximum, mean, and standard deviation (SD). All but two proteins (*KIF3C*; *PPL*) were more abundant in EC cases than in controls.

In Appendix A, we list all 151 variables whose values completely separate high-grade EC cases from low-grade EC cases.

Overlapping the 42 proteins separating CTRLs from ECs and the 151 separating high-grade from low-grade ECs, we found two proteins able to completely separate the three groups: *PRRC2A* and *SYDE2*. Their values for the three groups are reported in Appendix A, with corresponding box-plots in Figure 4 and Figure 5. For these two proteins, the abundances of CTRLs is lower than both high- and low-grade ECs. However, while for *PRRC2A*, high-grade ECs have higher values than low-grade ECs, for *SYDE2*, low-grade ECs have higher values than high-grade ECs.

The box represents the interquartile range (IQR), with the bottom and top edges at the 25th and 75th percentiles, respectively. The line in the middle of the box represents the median (50th percentile). The whiskers on a box-plot extend to the “adjacent values,” which are the lowest and highest data points within 1.5 times the interquartile range (IQR) from the box’s edges. Any data points beyond these adjacent values are considered outliers and are plotted individually.

To visually represent the overall distribution of protein expression changes and statistical significance across the three comparisons, volcano plots were generated (Figure 6). These plots highlight the differentially expressed proteins between HG vs. LG, high EC vs. CTRL, and low EC vs. CTRL groups.

The two sets of volcano plots originate from different analytical procedures. The first set of volcano plots were generated from the global differential expression analysis, based on fold change thresholds (≥1.5 or ≤0.6) combined with the Mann–Whitney U test (*p* < 0.05), and illustrate the overall distribution of significantly up- and downregulated proteins across the compared groups. The second set of volcano plots instead derive from the statistical analysis performed for biomarker identification. In this case, the plots highlight the proteins that completely discriminate between groups according to the applied statistical criteria. Therefore, while the first volcano plots depict the general pattern of proteomic deregulation, the second focuses on proteins with the highest discriminatory power.

### 2.3. Bioinformatic Analysis of Identified Proteins

Subsequently, we conducted a bioinformatic analysis using the g:Profiler tool to investigate the functional significance of all the proteins identified through analysis. The bioinformatics tool was employed to characterize the proteins according to their molecular functions, involvement in biological processes, and association with cellular components (Figure 7).

In terms of molecular function, proteins were ranked into purine nucleotide binding, integrin binding, extracellular matrix structural constituent, enzyme binding, and ATP binding. In terms of biological processes, the proteins were categorized into supramolecular fiber organization, cytoskeleton organization, cell migration, cell motility, protein metabolism, and protein folding. Furthermore, in terms of cellular components, they were categorized into extracellular exosomes, extracellular membrane-bounded organelles, cell junction, anchoring junction, extracellular space, and collagen-containing extracellular matrix. Pathway analysis using the Reactome database revealed that the differentially abundant proteins are primarily involved in neutrophil degranulation, the formation of a pool of free 40S subunits, the cross-presentation of soluble exogenous antigens, and the regulation of ornithine decarboxylase (ODC) (Figure 8).

IPA identified 68 dysregulated proteins significantly associated with “formation of focal adhesions”. A functional network was generated to illustrate their interactions (Figure 9).

### 2.4. Mass-Spectrometry Validation by Western Blotting in an Independent Cohort

To broaden the relevance of our proteomic findings, we validated the abundance of protein *SYDE2* and *ARPC2* (Figure 10) in an independent patient sample set consisting of 19 CTRL, 19 EC type 1 low-grade, 19 EC type 1 high-grade by Western blot analysis (Appendix A). For the validation, we selected these proteins because were able to separate cases from controls. Both of them were up-regulated in EC compared to the controls. Unfortunately, we could not validate PRRC2A because the antibody for this protein is not available commercially. The Mann–Whitney sum-rank test on *SYDE2* showed a statistically significant abundance (*p* < 0.05) in EC low-grade vs. CTRL (*p* = 0.048) and EC high-grade vs. CTRL (*p* = 0.001). The same test on *ARPC2* showed a statistically significant abundance (*p* < 0.05) in EC low-grade vs. CTRL (*p* = 0.005) and EC high-grade vs. CTRL (*p* = 0.0092).

## 3. Discussion

Cancer biomarkers are crucial for identifying tumor-specific changes and are widely used for disease diagnosis, prognosis, and guiding personalized treatment strategies [13]. Despite numerous studies that have been conducted to identify EC biomarkers, no protein has yet reached clinical application as biomarker. Many studies focus on biomarker discovery, while fewer address their validation [14].

Biomarker research in serum or plasma is particularly challenging due to the high dynamic range of protein concentrations and the typically low abundance of tumor-derived proteins exploitable as potential biomarkers [15]. The study of biofluids may overcome these limitations, deriving directly from the tumor tissue and bypassing abundant plasma proteins [16]. Our study is the first proteomic analysis of endometrial cancer tissue washings, exploring their potential as a protein-rich sample source of biomarkers, as demonstrated by the identification of over 4000 non-plasma cellular proteins in these samples.

In a cohort of EC patients, we analyzed the proteome of washing fluid by a discovery-based DIA approach, discovering distinct proteomic signatures that clearly differentiate EC patients from controls. Unlike absolute quantification techniques (e.g., PRM/SRM or ELISA), the standard curves are not applicable to this untargeted approach. Thus, to ensure reliability, we validated selected proteins using orthogonal techniques (Western blot), reinforcing the robustness of our findings.

Through univariate analysis we identified 42 proteins that are able to distinguish between patients and healthy controls. Furthermore, we identified 152 proteins capable of completely distinguishing type 1 EC patients with low-grade disease from patients with high-grade disease. Notably, among the proteins identified in the initial screening, *PRRC2A* and *SYDE2* were able to completely discriminate the three groups (controls, low-grade, and high-grade type 1 EC).

This finding highlights their strong biomarker potential and suggests that these proteins alone could support strong diagnostic potential without requiring complex multivariate models.

Our results analysis indicates that tissue washings are a highly effective source of biomarkers, as they allow the identification of soluble tissue proteins able to discriminate between controls and EC patients with much greater efficiency than serum [8,17].

Bioinformatic analysis with the g:Profiler tool provided insights into the functional annotation, revealing that the differentially abundant proteins are enriched in pathways related to protein metabolism, cytoskeleton remodeling, and cell motility, known processes central to tumor progression and metastasis. These findings suggest that the dysregulated proteins identified in our study are not only markers of disease presence but also may contribute to the invasive behavior of EC, representing potential targets for further investigation [18].

Protein metabolism, encompassing both the synthesis and degradation of proteins, is profoundly altered in cancer. This dysregulation plays a pivotal role in supporting the high proliferative and metabolic demands of tumor cells [19]. Cancer cells typically enhance protein synthesis to sustain rapid growth and maintain cellular functions essential for proliferation. At the same time, they activate catabolic pathways, including proteolysis, to recycle intracellular components and ensure a steady supply of amino acids, particularly under nutrient-limited or stress conditions [20]. In this context, amino acid metabolism also undergoes significant reprogramming. The enrichment of protein metabolism-related pathways observed in our study reinforces their importance in EC biology and suggests that targeting specific nodes of these processes may offer novel therapeutic strategies [21].

Further analysis with gProfiler software 1.0 (https://biit.cs.ut.ee/gprofiler/gost, accessed on 18 November 2025) showed that the differentially enriched proteins belong to exosomal and extracellular matrix components, both of which represent key elements for tumor-microenvironment communications that, in turn, sustain tumor growth, invasion, and metastasis [22]. Tumor-derived exosomes, through their cargo, can remodel the ECM, enhance intercellular communication, and promote cancer progression [23]. Likewise, changes in the extracellular matrix’s composition have a direct impact on cancer cell functions, including proliferation, survival, migration, and differentiation [24].

These findings highlight the importance of the ECM and exosome-associated proteins as players in EC biology and warrant further investigation.

Moreover, Reactome pathway analysis confirmed dysregulation of the ECM-related pathway, and highlight the role of *MECP2* and *SLC2A4*.

Briefly, *MECP2*, a methyl-CpG-binding protein, is a well-known epigenetic regulator with an oncogenic role, implicated in tumor cell proliferation, metastasis, and epithelial–mesenchymal transition not only by gene regulation but also through protein interactions [25,26].

The *SLC2A4* gene encodes for the glucose transport *GLUT4*, whose membrane translocation is regulated by insulin signaling. While its role in endometrial cancer is still elusive, evidence from breast cancer suggests that it may be involved in cellular transport and insulin signaling [27]. These pathways provide additional clues into how EC cells adapt their microenvironment and metabolism to support progression.

Ingenuity Pathway Analysis (IPA) showed that dysregulated proteins originated from diverse cellular compartments, underscoring the complexity of the EC proteome and the richness of tissue washings as a sample source. Within this network, three proteins emerged as particularly relevant: *PRRC2A*, *ARPC2*, and *SYDE2*, briefly described below. PRRC2A is a protein involved in RNA metabolism and protein translation initiation, frequently upregulated in various cancers [28]. It promotes tumor progression through activation of oncogenic signaling pathways, including *WNT* and *YAP* [29], and may facilitate tumor immune evasion by inhibiting immune cell infiltration and promoting T cell exhaustion [28].

*ARPC2* represents a subunit of the Arp2/3 complex, controls actin polymerization, and modulates cytoskeletal dynamics favoring cancer cells migration, adhesion, and invasion [30]. It has been associated with poor clinical outcomes in gastric and breast cancers by these processes, and with modulating the proliferative capacity of cancer cells [31,32].

*SYDE2* encodes a *Rho GTP*ase-activating protein whose role in tumorigenesis remains to be fully elucidated. However, evidence from cell renal cell carcinoma (ccRCC) suggest it correlates with immune infiltration and may have context-dependent effects as a tumor-suppressor [33,34].

Together, these proteins not only serve as potential biomarkers but also provide mechanistic insight into pathways driving EC progression.

Collectively, this study provides a comprehensive proteomic characterization of EC tissue washings, and identifies novel candidate biomarkers with potential clinical relevance. Our findings establish tissue washings as a rich and clinically informative source of tumor-derived proteins, alternative to serum or plasma for biomarker discovery. The translation of such biomarkers into clinical practice could accelerate the diagnostic process and reduce healthcare costs. In this context, tissue washings represent a promising alternative to serum or plasma, offering higher specificity for the identification of EC-associated biomarkers [35].

Nevertheless, several limitations must be acknowledged. The absence of a suitable antibody prevented the validation of PRRC2A, despite its strong biomarker potential. In addition, while our validation cohort was independent, it remained relatively small. Larger, multicenter studies are required to confirm the generalizability and clinical relevance of these findings. Future work should focus on developing targeted assays (e.g., PRM/SRM or ELISA) for candidate proteins, integrating proteomic findings with genomic and transcriptomic data, and investigating the functional role of key proteins such as SYDE2 and PRRC2A in EC progression. With continued validation, these proteins could serve as valuable diagnostic and prognostic biomarkers and potentially guide personalized treatment strategies.

## 4. Materials and Methods

### 4.1. Patients

During 2014 and 2024, a total of 131 patients were enrolled at the Institute for Maternal and Child Health—IRCCS “Burlo Garofolo” in Trieste, Italy. This included 87 women diagnosed with EC and 44 CTRL subjects. The study was approved by the Institute’s Technical and Scientific Committee and all procedures were conducted in accordance with the principles of the Declaration of Helsinki. Written informed consent was obtained from all participants prior to inclusion. The median age of patients with EC type 1 low-grade was 73.5 years (min = 52, max = 95), while the median age of patients with EC type 1 high-grade was 71.5 years (min = 49, max = 93). The median age of the controls was 44.5 years (min = 30, max = 75). All patients underwent a physical examination and transvaginal ultrasound as part of the screening process for gynecological pathologies. Tumor tissue and corresponding control samples were collected immediately following surgery. The inclusion criteria comprised patients aged over 18 years in both groups. For the patient group, this included individuals diagnosed with EC confirmed by post-hysteroscopic histological examination (FIGO stages I–IV). Like a control group, patients with leiomyoma, with non-endometrial pathology, or with negative endometrial histology were included. The exclusion criteria for both groups exclude patients with infectious diseases (e.g., HIV, HBV, or HCV), adenomyosis, the presence of synchronous tumors, as well as individuals with a history of other neoplastic conditions and/or those who had undergone chemotherapy or radiotherapy within the past 10 years.

### 4.2. Sample Collection

Having obtained the biopsy for EC and CTRL, it was washed from the blood and put in 10 mL tubes with 5 mL NaCl 0.9%. The tubes were centrifuged at 16,000× *g* for 30 min at 4 °C to remove cell debris. After centrifugation, 5 mL of supernatant was desalted and concentrated using a Vivaspin Concentrator Filter Unit (EMD Millipore, Billerica, MA, USA) with a molecular weight cut-off of 3 kDa at 4000× *g* at 25 °C until the remaining volume reached 50–100 μL. Protein content was determined using Bradford reagent.

### 4.3. Protein Digestion and MS Analysis

A total of 20 µg of protein was subjected to enzymatic digestion using the EasyPep™ MS Sample Prep Kit (Thermo Fisher Scientific Waltham, MA, USA). Following digestion, peptide analysis was carried out using nUHPLC-HRMS. Analyses were performed on an Ultimate 3000 nanoLC system (Thermo Fisher Scientific, Bremen, Germany) interfaced with an Orbitrap Lumos Tribrid mass spectrometer (Thermo Fisher Scientific) equipped with an Easyspray nESI source (Thermo Fisher Scientific). An initial injection volume of 1 μL of the digested sample was loaded onto a PepMap trap column and retained for 1.0 min at a flow rate of 60 μL/min. Peptides were subsequently separated on a C18 reversed-phase analytical column (250 mm × 75 μm inner diameter, 2.0 µm particle size, 100 Å pore size; EasySpray PepMap, Thermo Scientific).

A linear 60 min gradient was performed. For data-independent acquisition (DIA), an initial full MS scan was acquired at a resolution of 60,000. Subsequent DIA scans were performed in the Orbitrap at a resolution of 30,000, using 10 Da isolation windows. The automatic gain control (AGC) target and maximum ion injection time were set to custom values of 200 ms and 40 ms, respectively. The loop count was set to 30. Higher-energy collisional dissociation (HCD) was employed with a normalized collision energy of 30.

DIA raw data were processed using Spectronaut version 19, employing the DirectDIA (deep) workflow for peptide and protein identification. Carbamidomethylation was set as a fixed modification, while acetylation (protein N-terminus) and methionine oxidation were specified as variable modifications. A false discovery rate (FDR) threshold of <1% was applied at both the peptide and protein levels. Proteins were considered confidently identified if supported by at least one unique peptide. The software enables relative label-free quantification based on peptide ion intensities matched to a spectral library. Data were normalized across runs using Spectronaut’s built-in global normalization algorithm. Retention time alignment was achieved using a non-linear regression model based on indexed Retention Time (iRT) peptides, ensuring high precision in peptide identification and quantification across LC-MS runs. To ensure high-confidence quantification, we excluded proteins with missing intensity values. The mass spectrometry proteomics data have been deposited into the ProteomeXchange Consortium via the PRIDE partner repository with the dataset identifier PXD069126.

### 4.4. Western Blotting

Western blot analysis of tissue washing was conducted as previously described [36]. For this procedure, 30 µg of total protein was loaded onto a 4–20% precast polyacrylamide gel and subsequently transferred onto a nitrocellulose membrane. The membrane was then blocked with 5% non-fat dry milk in TBS-Tween 20 (TBS-T) and incubated overnight at 4 °C with primary antibodies against SYDE2 (1:500, rabbit polyclonal) and ARPC2 (1:1000, rabbit polyclonal) (Thermo Fisher Waltham, MA, USA). Following primary antibody incubation, membranes were treated with horseradish peroxidase (HRP)-conjugated secondary antibodies (anti-rabbit IgG, 1:3000; Sigma-Aldrich, St. Louis, MO, USA; Merck KGaA, Darmstadt, Germany). Protein bands were visualized using the SuperSignal West Pico Chemiluminescent substrate. Band intensities were quantified by normalizing to the total protein content, as determined by Ponceau Red staining of the same blot membrane.

### 4.5. Bioinformatic Analysis

Proteins identified through MS were analyzed with the gProfiler classification systems, and categorized according to their molecular function involvement, biological processes, and protein class. Pathway analysis was carried out using the Reactome tool. The bio-functions were generated via Ingenuity Pathway Analysis (IPA) [37]. Results from the IPA were considered statistically significant when *p* < 0.01. For the filter summary, we only considered associations where the confidence was high (predicted), or for those that had been observed experimentally. Volcano plots, heatmaps, and principal component analysis (PCA) were generated using the MetaboAnalyst, SRplot platforms, and TBtools 2 software. (https://github.com/CJ-Chen/TBtools-II, accessed on 18 November 2025).

### 4.6. Statistical Approach

Having to deal with 4134 variables and 24 patients with high-grade EC type 1, 25 patients with low-grade EC type 1, and 25 controls might be extremely complicated if the objective is to find a model able to separate controls from EC patients and to separate the two types of EC. In addition, the dataset had an extremely high number of missing values due to the inability of the machine to return a numerical value in the cases in which values were too low. This lack of data could not be solved as we could not count on information about the numeric interval within which the machine could not return a numeric value.

In many variables with a significant amount of missing values, however, we noticed a pattern, i.e., the expression of specific variables was below detection level more frequently in one or two specific outcome groups.

In any case, given the impossibility of establishing the interval in which the machine could not return a numeric value and, consequently, unable to generate random numbers that could fill the gaps, we decided to start removing all variables with missing data. We were then left with 960 variables out of the original 4134.

We then ran a univariate logistic analysis with ECs vs. CTRLs as the dependent variable and each of the 960 variables as independent variables. This led to 42 variables completely separating ECs from controls. For these 42 variables, we calculated the median, the interquartile range, the range, the mean, and the standard deviation (SD). We also graphed the box-plots.

The same approach was adopted for the outcome advanced EC vs. low EC. Out of 960 variables, we found 151 which completely separated the two groups. Out of these 151, two were able to separate both ECs from controls and advanced ECs from low ECs PRRC2A; SYDE2).

Some of the results are anticipated in this section because the initial idea was to adopt a penalization approach to reduce the number of variables involved in modeling, with the objective being the identification of a model that could allow us to identify EC patients and separate advanced EC from low EC. However, this approach was overcome by the significant amount of missing data, and the exploratory finding of variables completely separating the three groups of interest, with no need for any statistical modeling.

## 5. Conclusions

For the first time in this study, we report a comprehensive proteomic analysis of tissue washings from endometrial cancer (EC) patients. This study demonstrates that endometrial cancer tissue washings are a rich source of tumor-derived proteins, outperforming serum in biomarker discovery. Notably, the validated proteins allow us to distinguish low- and high-grade type 1 EC patients from healthy controls, suggesting that its translation into routine practice could reinforce EC management strategies. From over 4000 identified proteins, we highlight *PRRC2A*, *SYDE2*, and *ARPC2* as promising candidates for diagnosis and disease stratification, with *SYDE2* and *ARPC2* validated in an independent cohort. Functional and pathway analyses reveal dysregulation of protein metabolism, cytoskeletal remodeling, ECM organization, and exosome biology, processes that drive tumor progression and metastasis. With further validation in larger cohorts and the development of targeted assays, these biomarkers could advance early detection, improve risk stratification, and support personalized treatment strategies in EC.

## Figures and Tables

**Figure 1 ijms-26-11498-f001:**
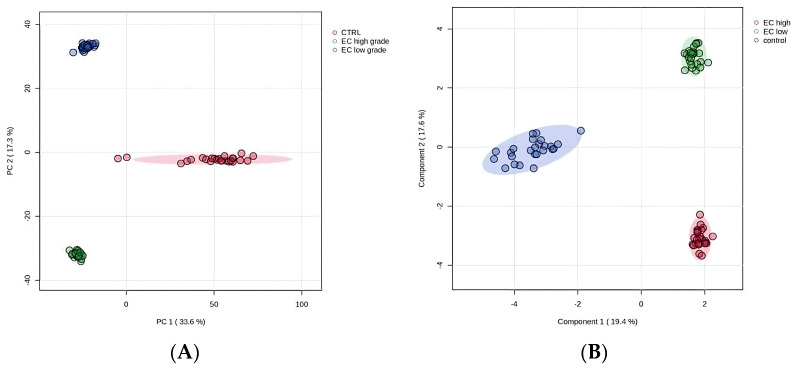
(**A**) PCA plot shows the distribution of samples based on the first two principal components (PC1 and PC2), explaining 33.6% and 17.3% of the variance, respectively. Ellipses denote 95% confidence intervals for each group: EC high (red), EC low (green), and control (blue). (**B**) PLS-DA score plot illustrating enhanced separation of the three groups.

**Figure 2 ijms-26-11498-f002:**
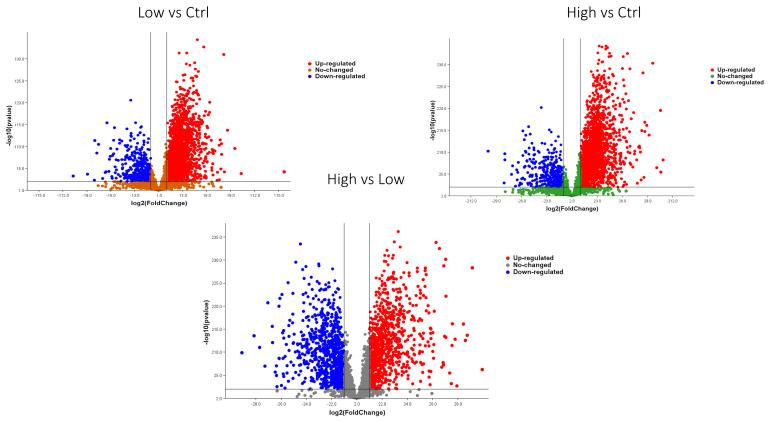
Volcano plots representing the results of the statistical analysis for the three comparisons. Statistical significance was determined using the Mann–Whitney U test (*p* < 0.05), combined with fold change thresholds (≥1.5 or ≤0.6). Proteins with fold changes ≥ 1.5 or ≤0.6 are highlighted to indicate significant upregulation or downregulation, respectively. High (high-grade), Low (low-grade).

**Figure 3 ijms-26-11498-f003:**
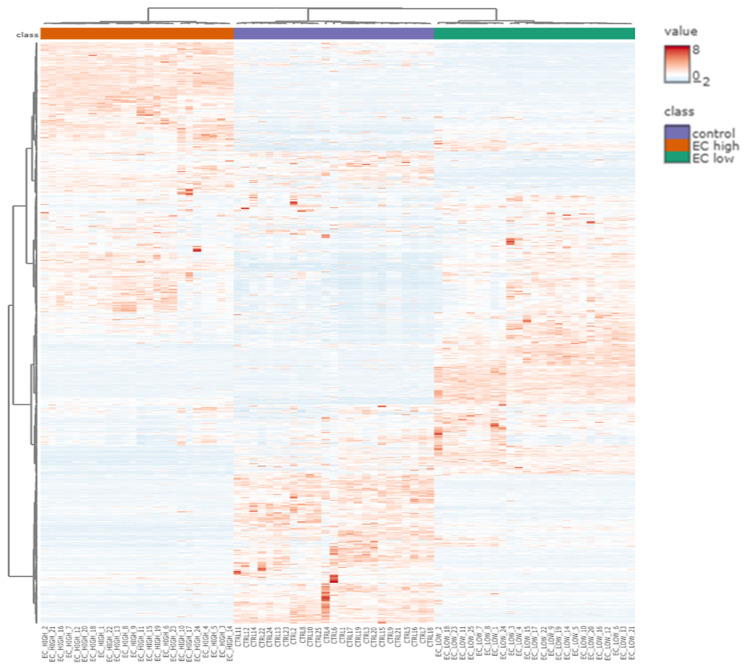
Hierarchical clustering heatmap of the 3661 dysregulated proteins identified across the three experimental groups: control (blue), endometrial cancer (EC) low-grade (green), and EC high-grade (orange). Rows represent individual proteins, while columns correspond to samples. Protein abundance values are represented by a color gradient from blue (low expression) to red (high expression). The dendrograms illustrate clustering patterns among proteins and samples.

**Figure 4 ijms-26-11498-f004:**
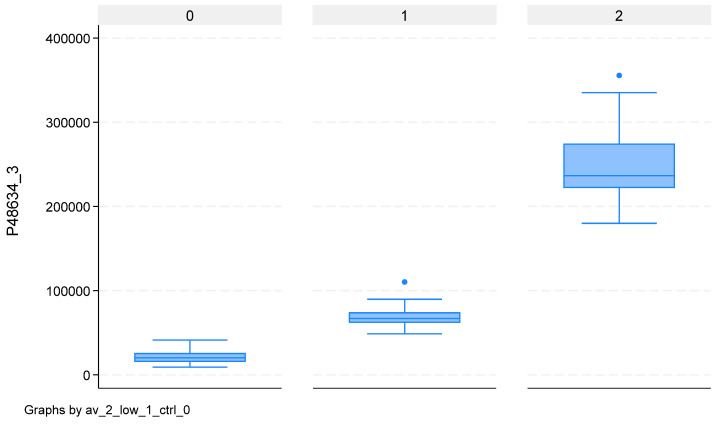
Box-plot of abundance values for protein PRRC2A for controls (0), low-grade EC cases (1), and high-grade EC cases (2).

**Figure 5 ijms-26-11498-f005:**
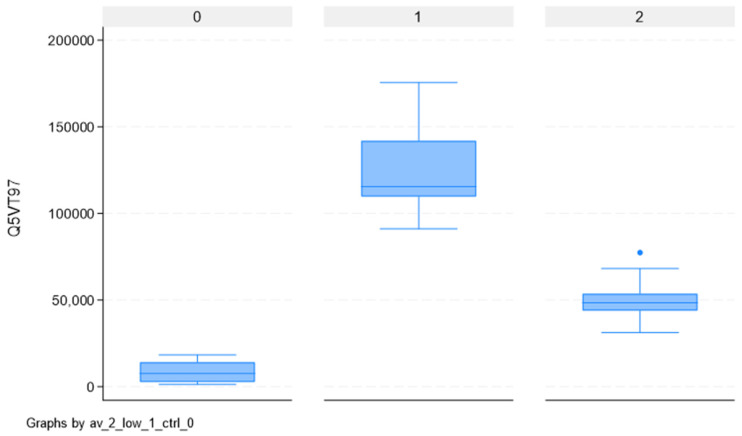
Box-plot of abundance values for protein SYDE2 for controls (0), low-grade EC cases (1), and high-grade EC cases (2).

**Figure 6 ijms-26-11498-f006:**
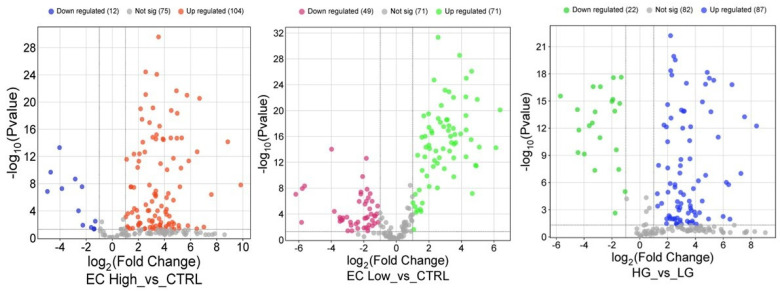
Volcano plots illustrating the distribution of differential protein expression across the three group comparisons. A total of 192 proteins showed significant changes based on the Mann–Whitney U test (*p* < 0.05) combined with fold change thresholds (≥1.5 for upregulation and ≤0.6 for downregulation). Proteins meeting both criteria are highlighted to indicate statistically significant upregulation or downregulation. Comparisons include high-grade endometrial cancer (HG) vs. control, low-grade endometrial cancer (LG) vs. control, and HG vs. LG.

**Figure 7 ijms-26-11498-f007:**
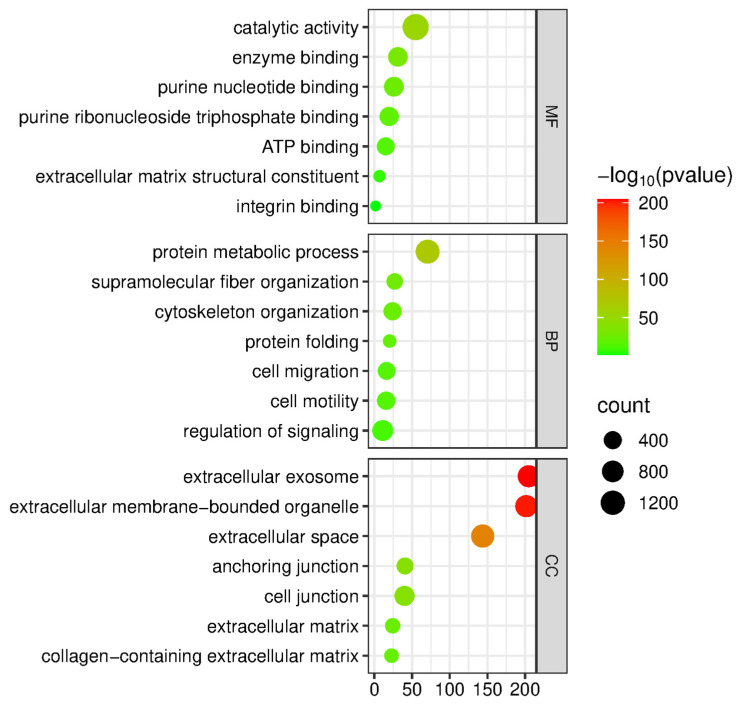
Functional enrichment analysis of 3661 dysregulated proteins using g:Profiler. Bubble plots show combined GO enrichment across biological processes, molecular functions, and cellular components. Bubble size represents the number of proteins per term, while color intensity reflects the significance level as −log10(*p*-value). This visualization highlights key pathways, functions, and cellular locations associated with the dysregulated proteins. The *x*-axis indicates the number of dysregulated proteins associated with each GO term (i.e., term count).

**Figure 8 ijms-26-11498-f008:**
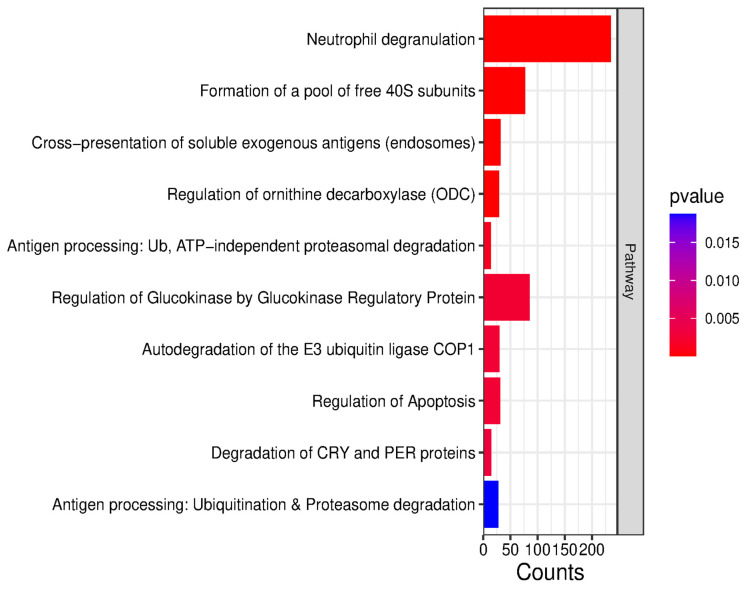
Reactome pathway enrichment analysis based on all differentially abundant proteins. Bar lengths indicate the number of proteins associated with each pathway (Counts), while the color gradient reflects statistical significance (*p*-value).

**Figure 9 ijms-26-11498-f009:**
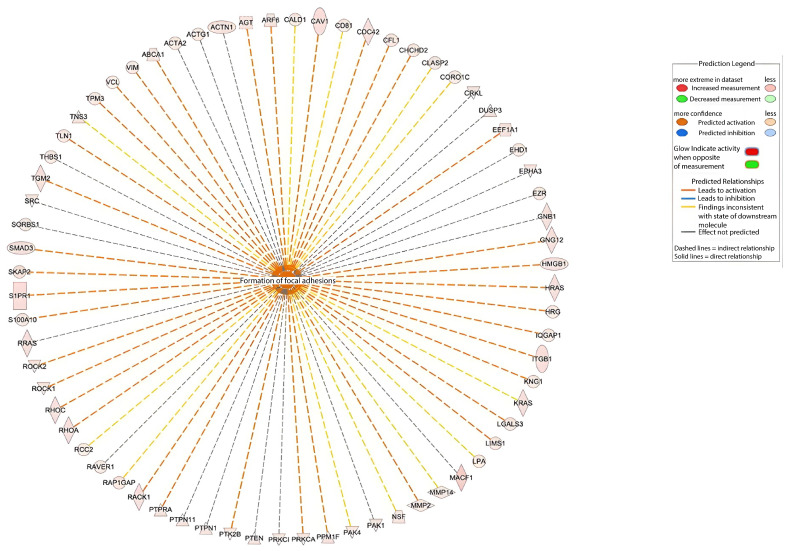
IPA-derived network of proteins linked to focal adhesion formation. Nodes represent dysregulated proteins; edges indicate predicted relationships (solid = direct, dashed = indirect). Central gray node = “formation of focal adhesions” biofunction.

**Figure 10 ijms-26-11498-f010:**
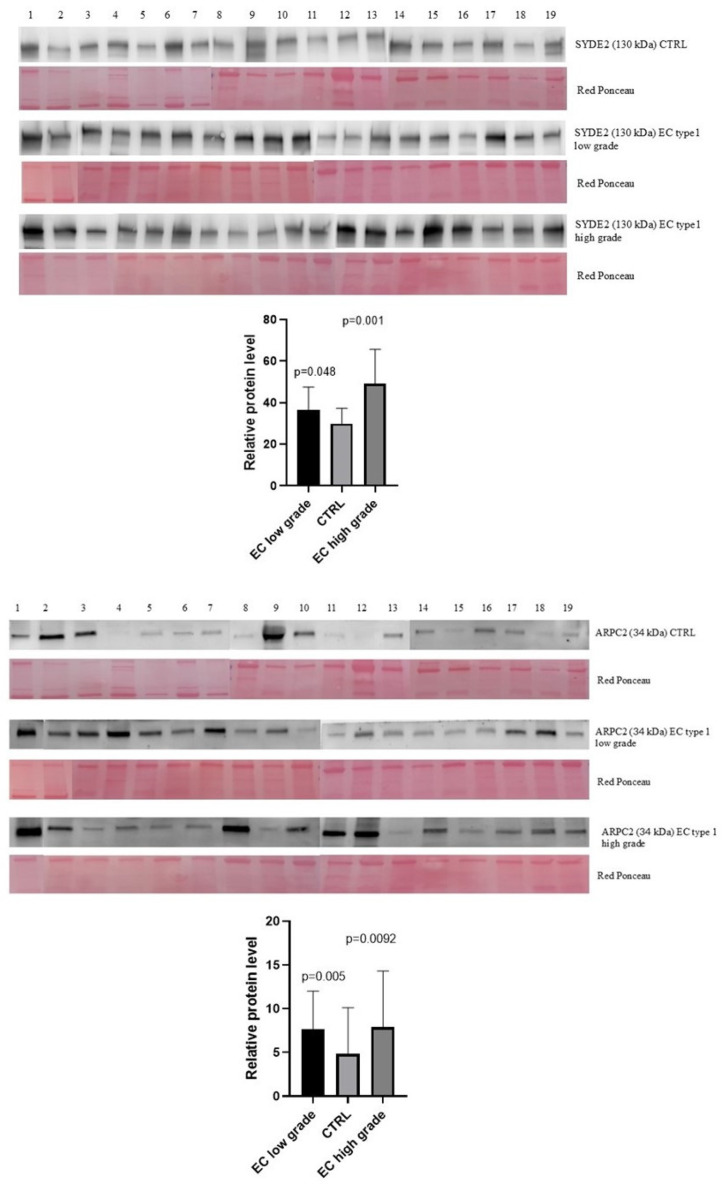
The expression levels of *SYDE2* and *ARPC2* proteins in endometrial cancer (EC) low-grade, EC high-grade, and normal endometrial tissue (CTRL) were assessed by Western blot analysis. Band intensities of immunodetected proteins were normalized to total protein levels, as visualized by Ponceau Red staining of the same blot. Quantitative results are presented as a histogram, with each bar representing the mean ± standard deviation. Statistical significance was determined at *p* < 0.05.

## Data Availability

The original contributions presented in this study are included in the article/Appendix A. Further inquiries can be directed to the corresponding author.

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
