# Peer review of "Data-Independent Acquisition (DIA)-Based Proteomics for the Identification of Biomarkers in Tissue Washings of Endometrial Cancer"

_ijms, 2025, doi:10.3390/ijms262311498_

Round 1
Reviewer 1 Report
Comments and Suggestions for Authors
The manuscript presents proteomics data relevant to endometrial cancer diagnosis by comparing control tissue washings with low-grade (LG) and high-grade (HG) tumour samples. The mass spectrometry platform used, and the DIA method are appropriate for an in-depth characterisation of the endometrial washings. Some of the proteomics findings were validated by WB in an independent sample cohort.
The following points need to addressed to improve the quality of the manuscript:
1) The original dataset has 4134 proteins identified. The authors removed all proteins with missing intensity values and retained only 960 proteins supposedly with available intensities in all samples. This method is not acceptable. A standard procedure is to keep all proteins with values in at least 40% of the samples and use Spectronaut or DIA-NN for imputation of missing values. The increased number of retained proteins will offer improved insights on the molecular processes associated with disease onset and progression, and more biomarker candidates.
2) It is necessary to add in a supplementary file the list of all identified proteins and their quantitation data in all the samples. This file should also contain the mean and standard deviation for all the proteins in the three groups (control, LG, HG). The raw data must be deposited in the PRIDE database and thus made publicly available.
3) All Figures must have a title and an informative legend allowing the reader to understand and interpret the data presented.
4) Tables 1 and 2 should be moved to supplementary information and replaced with 2 Volcano plots for comparisons control vs cases and LG vs HG, and one heat map presenting all the differentially expressed proteins.
5) A Principal Component Analysis (PCA) figure should be added to illustrate the grouping of the different samples.
6) It is recommended to use ELISA instead of WB for validation of the proteomics results since ELISA can be easily implemented in a clinical lab.
Author Response
Reviewer 1
The original dataset has 4134 proteins identified. The authors removed all proteins with missing intensity values and retained only 960 proteins supposedly with available intensities in all samples. This method is not acceptable. A standard procedure is to keep all proteins with values in at least 40% of the samples and use Spectronaut or DIA-NN for imputation of missing values. The increased number of retained proteins will offer improved insights on the molecular processes associated with disease onset and progression, and more biomarker candidates.
Our reply: We thank the reviewer for the valuable suggestion to retain proteins with intensity values available in at least 40% of the samples and to consider missing value imputation, which is a common approach in proteomics analyses using tools such as Spectronaut or DIA-NN.
However, in our study we employed Spectronaut 19, which does not natively support missing value imputation. Therefore, we opted to remove proteins with missing intensity values to ensure that our dataset contains only quantifications based on confident peptide evidence.
While we recognize that imputing missing data can enhance proteome coverage and potentially identify additional biomarker candidates, this process may also introduce artificial signals that are not directly supported by the measured peptide intensities.
Given that missing values in data-independent acquisition (DIA) are often due to peptides being below detection limits, imputing such values could risk generating less reliable quantifications. Thus, we prioritized data reliability and robustness by focusing on proteins with complete observed intensity data for downstream analyses and biomarker discovery.
In addition, while we agree that among the excluded incomplete proteins some might be promising, we have enough information (even without artificial imputations) to develop a predictive model that completely separates the three groups.
It is necessary to add in a supplementary file the list of all identified proteins and their quantitation data in all the samples. This file should also contain the mean and standard deviation for all the proteins in the three groups (control, LG, HG). The raw data must be deposited in the PRIDE database and thus made publicly available.
Our reply: We added the suggested analysis in a supplementary file. However, given the high number of missing values due to values below LOD, we refrained from calculating the statistics for those groups and proteins in which we had more than 40% missing values. This is also due to the fact that the specific protein LOD is unknown.. The mass spectrometry proteomics data have been deposited to the ProteomeXchange Consortium via the PRIDE partner repository with the dataset identifier PXD069126.
All Figures must have a title and an informative legend allowing the reader to understand and interpret the data presented.
Our reply: We fixed the figures.
Tables 1 and 2 should be moved to supplementary information and replaced with 2 Volcano plots for comparisons control vs cases and LG vs HG, and one heat map presenting all the differentially expressed proteins.
Our reply: As suggested by the reviewer, we have moved Tables 1 and 2 to the Supplementary Information. We have also added volcano plots comparing EC High vs CTRL, EC Low vs CTRL, and EC High vs EC Low. In addition, we have included a heat map presenting all the differentially expressed proteins.
A Principal Component Analysis (PCA) figure should be added to illustrate the grouping of the different samples.
Our reply: In accordance with the reviewer’s suggestion, we have added both PCA and PLS-DA analyses to better illustrate the grouping of the different samples.
It is recommended to use ELISA instead of WB for validation of the proteomics results since ELISA can be easily implemented in a clinical lab.
Our reply: We thank the reviewer for this valuable suggestion. While we agree that ELISA is more widely adopted in clinical practice, the available ELISA kits for PRRC2A and SYDE2 have not yet been validated, and such assays typically require extensive optimization to ensure reliability. For this study, we therefore selected Western blotting as a more robust and well-established approach, using antibodies that have already been validated and successfully applied in previous experiments.
Reviewer 2 Report
Comments and Suggestions for Authors
This study presents a typical set of problems often encountered in omics analyses, primarily due to insufficient measurement validation and inadequate data analysis.
Specific Comments:
- Validation of Measurement Methodology: The study claims to be the first to perform proteomic analysis of endometrial cancer tissue washings using data-independent acquisition (DIA) mass spectrometry. However, the validity and reliability of the DIA-MS measurement itself have not been adequately verified. While Western blot validation was performed for two biomarkers, the fundamental validity of the DIA-based analysis is not established. This raises concerns that essential biomarkers might have been overlooked. Ideally, a standard curve should first be generated using several proteins at different concentrations to confirm linearity and the dynamic range of detection. The authors should also confirm that actual sample concentrations fall within this linear range, thus initially evaluating the relationship between the data signal and concentration.
- Normalization of DIA Data: While normalization is performed for Western blot analysis, there is no mention of normalization for the DIA data. If there are variations in the overall concentration (e.g., "thick" or "thin" samples) within the tissue washing fluid, the current data analysis method may not have removed this influence. For example, in urine analysis, creatinine is often used for data correction; a similar consideration may be necessary here. To assess the necessity of such normalization, the authors should visualize as many variables as possible (e.g., using heatmaps) to check for overall concentration differences across samples. Principal component analysis (PCA) or similar methods should also be used to identify potential outliers at the sample level. Furthermore, before calculating final statistical values, it is crucial to confirm that overall concentration differences do not impact results, perhaps by examining volcano plots that include statistically non-significant substances. The current approach is inadequate for omics analysis, as it observes numerous variables but relies on conventional statistics, potentially overlooking these critical issues.
- Handling of Missing Values: A significant number of variables contained missing values, but the authors chose to delete these variables rather than addressing the underlying problem. This approach also risks overlooking many potentially essential biomarkers.
- Comprehensive Data Analysis and Focus: Although a large amount of data was measured, the discussion primarily revolves around the two final biomarkers (PRRC2A and SYDE2). It would be more appropriate to comprehensively analyze the entire dataset—not just the selected biomarkers and their direct associations—and then gradually narrow the focus to the most promising candidates. While bioinformatic analyses were performed on a subset of statistically identified proteins, the authors explicitly stated that they did not conduct further statistical modeling or multivariate analysis because univariate analysis identified variables capable of "completely separating the three groups". This decision may have limited the exploration of complex biological interactions and broader mechanisms within the extensive proteomic dataset. A more thorough analysis, including multivariate approaches and clustering of the entire dataset, would yield richer scientific insights and provide a more robust basis for selecting and interpreting key biomarkers.
- Table Presentation: Tables 1 and 2 are excessively large and redundant for inclusion in the main manuscript. They should be moved to Supplementary Information. Table 3, though not as large, should also be treated similarly.
- Patient Information and Confounding Factors: Before presenting protein tables, patient demographic information should be summarized in a table with appropriate statistical values to compare groups. Notably, there is a significant age difference between the control group (median age 44.5 years) and the EC groups (low-grade 73.5 years, high-grade 71.5 years). If statistically significant differences exist in patient characteristics, these should be addressed as confounding factors, potentially through multivariate analysis in conjunction with the identified biomarkers. The lack of such consideration undermines the robustness of the findings.
- Insufficient Figure Captions (Figures 1 & 2): The captions for the Box plots (Figures 1 and 2) are overly simplistic. They lack essential information such as the definition of the box's horizontal lines (e.g., median), the definition of the whiskers, the statistical test used for comparison, and the meaning of any asterisks or p-values indicating statistical significance. This omission hinders the reader's ability to understand the data presented entirely.
- Insufficient Figure Captions (Figure 3): The caption for Figure 3 is insufficient. It requires a detailed explanation of what the size and color of the plots represent, as well as the meaning of the X-axis, to correctly interpret the bioinformatics enrichment analysis.
- Insufficient Figure Captions (Figures 4 & 5): Similarly, Figures 4 and 5 need comprehensive explanations for the color and size of nodes and edges, and any other visual elements, to allow for proper interpretation of the Reactome pathways and Ingenuity Pathway Analysis network.
- Clarity in Three-Group Comparisons: When performing three-group comparisons, it must be clearly specified which groups are being compared (e.g., CTRL vs. low-grade EC, low-grade EC vs. high-grade EC, CTRL vs. high-grade EC). The methods for statistical comparison for each pair, including the specific tests and p-values, should be explicitly stated.
Author Response
Reviewer 2
Validation of Measurement Methodology: The study claims to be the first to perform proteomic analysis of endometrial cancer tissue washings using data-independent acquisition (DIA) mass spectrometry. However, the validity and reliability of the DIA-MS measurement itself have not been adequately verified. While Western blot validation was performed for two biomarkers, the fundamental validity of the DIA-based analysis is not established. This raises concerns that essential biomarkers might have been overlooked. Ideally, a standard curve should first be generated using several proteins at different concentrations to confirm linearity and the dynamic range of detection. The authors should also confirm that actual sample concentrations fall within this linear range, thus initially evaluating the relationship between the data signal and concentration.
Our reply: We thank the reviewer for raising this point regarding the validation of our DIA-MS–based proteomic approach. We would like to clarify that our study employs a shotgun proteomics workflow using Data-Independent Acquisition (DIA), which is a discovery-driven, label-free method that provides relative quantification based on peptides. In contrast to targeted or absolute quantification approaches (e.g., ELISA or PRM/SRM with isotope-labeled standards), DIA-MS does not rely on external calibration curves for quantification. Instead, relative protein abundance is inferred through spectral library–guided extraction and integration of peptide fragment ion signals—a well-established and widely accepted strategy in exploratory proteomics.. While we appreciate the reviewer’s suggestion, the generation of a standard curve is not a routine or appropriate validation method in untargeted bottom-up DIA workflows. Moreover, in the specific context of uterine lavage fluid, there are no well-characterized endogenous reference proteins that could serve as internal or external standards. The protein content in this sample type is highly variable across individuals, which makes the construction of a meaningful and representative calibration curve both technically unfeasible and biologically uninformative. To ensure the robustness and reproducibility of our results, we applied stringent quality control, normalization, and statistical filtering throughout the DIA-MS data analysis. Additionally, we validated two of the differentially expressed proteins using Western blotting, which confirmed the expression trends observed in the DIA-MS data and provided orthogonal support for our findings. We have now included this methodological clarification in the revised manuscript. We believe this addition will help readers better understand the strengths and appropriate applications of the DIA-MS approach in complex biological matrices. Furthermore, (Figure.1) has been added to illustrate the non-linear retention time calibration performed in Spectronaut®, based on iRT peptides. This correction model ensures accurate alignment of peptide retention times across runs, thus improving the precision and reliability of identification and quantification in the DIA workflow.
Figure 1. Non-linear retention time calibration in Spectronaut®. The plot shows the relationship between observed retention time (RT) and indexed retention time (iRT) peptides across the LC gradient. The black curve represents the non-linear regression model used to align retention times, enabling accurate peptide identification and quantification in DIA-MS analysis.
References
- Bruderer, R., Bernhardt, O. M., Gandhi, T., Miladinović, S. M., Cheng, L. Y., Messner, S., ... & Reiter, L. (2017).Optimization of experimental parameters in data-independent mass spectrometry significantly increases depth and reproducibility of results.Molecular & Cellular Proteomics, 16(12), 2296–2309. https://doi.org/10.1074/mcp.RA117.000314.
- Gillet, L. C., Navarro, P., Tate, S., Röst, H., Selevsek, N., Reiter, L., Bonner, R., & Aebersold, R. (2012).Targeted data extraction of the MS/MS spectra generated by data-independent acquisition: A new concept for consistent and accurate proteome analysis. Molecular & Cellular Proteomics, 11(6), O111.016717. https://doi.org/10.1074/mcp.O111.016717.
Normalization of DIA Data: While normalization is performed for Western blot analysis, there is no mention of normalization for the DIA data. If there are variations in the overall concentration (e.g., "thick" or "thin" samples) within the tissue washing fluid, the current data analysis method may not have removed this influence. For example, in urine analysis, creatinine is often used for data correction; a similar consideration may be necessary here. To assess the necessity of such normalization, the authors should visualize as many variables as possible (e.g., using heatmaps) to check for overall concentration differences across samples. Principal component analysis (PCA) or similar methods should also be used to identify potential outliers at the sample level. Furthermore, before calculating final statistical values, it is crucial to confirm that overall concentration differences do not impact results, perhaps by examining volcano plots that include statistically non-significant substances. The current approach is inadequate for omics analysis, as it observes numerous variables but relies on conventional statistics, potentially overlooking these critical issues.
Our reply: We thank the reviewer for this valuable comment regarding normalization and quality control of the DIA data. We apologize for not clearly describing the normalization procedures in the original version of the manuscript. We would like to clarify that global normalization was performed using Spectronaut®, which applies normalization across runs based on global statistics (e.g., median, mean, or geometric mean of precursor intensities). This step corrects for systematic technical variation across LC-MS runs, including variability in total ion signal due to differences in sample handling or instrument performance. Unlike fluids such as urine, uterine lavage fluid does not have a well-established endogenous reference compound (e.g., creatinine) that could be used for absolute concentration correction. Protein content in uterine washings can vary considerably due to biological and procedural variability, and there is currently no consensus on a suitable internal standard for normalization in this matrix. For this reason, external concentration correction strategies (like creatinine normalization) are not applicable in our context. To address the reviewer’s concern regarding sample-to-sample variation, we now include exploratory data visualizations (e.g., heatmaps and principal component analysis [PCA]) to assess global variation and identify potential outliers. We also include volcano plots that display all quantified proteins, not only those reaching statistical significance, to better visualize the distribution of fold changes. These additions will help ensure that our conclusions are not confounded by technical or concentration-related variability, and we have clarified these points in the revised manuscript.
Handling of Missing Values: A significant number of variables contained missing values, but the authors chose to delete these variables rather than addressing the underlying problem. This approach also risks overlooking many potentially essential biomarkers.
Our reply: We thank the reviewer for raising the important point regarding the handling of missing values. Indeed, a considerable number of proteins in our dataset contained missing intensity values. In our study, we used Spectronaut 19, which, unlike some other proteomics software, does not provide native support for missing value imputation. Therefore, to ensure the highest confidence in our quantitative data, we made the conservative decision to exclude proteins with any missing intensity values across samples. This approach guarantees that all proteins retained in our analysis are supported by direct and confident peptide quantifications. We acknowledge that this method may reduce the total number of proteins analyzed and could potentially overlook biomarkers present in fewer samples. However, imputing missing values—especially when peptides are below detection limits—may introduce artificial data that could lead to false positives or biased interpretations. Our priority was to maintain data integrity and reliability, focusing on biomarkers substantiated by robust experimental evidence. We believe this conservative strategy provides a solid foundation for the biological conclusions drawn from our study. The information we have on complete proteins is more than enough to separate the three groups. We understand that several other proteins might be of interest, and it is our intention to further investigate the most promising ones.
Comprehensive Data Analysis and Focus: Although a large amount of data was measured, the discussion primarily revolves around the two final biomarkers (PRRC2A and SYDE2). It would be more appropriate to comprehensively analyze the entire dataset—not just the selected biomarkers and their direct associations—and then gradually narrow the focus to the most promising candidates. While bioinformatic analyses were performed on a subset of statistically identified proteins, the authors explicitly stated that they did not conduct further statistical modeling or multivariate analysis because univariate analysis identified variables capable of "completely separating the three groups". This decision may have limited the exploration of complex biological interactions and broader mechanisms within the extensive proteomic dataset. A more thorough analysis, including multivariate approaches and clustering of the entire dataset, would yield richer scientific insights and provide a more robust basis for selecting and interpreting key biomarkers.
Our reply: We thank the reviewer for the insightful suggestion. As recommended, we performed a comprehensive analysis of the entire set of proteins identified in the study, including additional enrichment analyses. Our univariate analysis identified 192 proteins that significantly distinguish high-grade and low-grade endometrial cancers from controls. Among these, PRRC2A and SYDE2 emerged as the most robust biomarkers, capable of completely separating the three groups. Given this clear separation, we considered additional multivariate modeling unnecessary, as these two proteins sufficiently discriminate cases from controls. We have expanded the Results and Discussion sections to include insights from the full proteomic dataset.
Table Presentation: Tables 1 and 2 are excessively large and redundant for inclusion in the main manuscript. They should be moved to Supplementary Information. Table 3, though not as large, should also be treated similarly.
Our reply: As suggested by the reviewer, the tables have been moved to the Supplementary Data section.
Patient Information and Confounding Factors: Before presenting protein tables, patient demographic information should be summarized in a table with appropriate statistical values to compare groups. Notably, there is a significant age difference between the control group (median age 44.5 years) and the EC groups (low-grade 73.5 years, high-grade 71.5 years). If statistically significant differences exist in patient characteristics, these should be addressed as confounding factors, potentially through multivariate analysis in conjunction with the identified biomarkers. The lack of such consideration undermines the robustness of the findings.
Our reply: We agree with the Reviewer that the age of patients can probably affect the abundance of proteins. However, we did not carry out any multivariate analysis, simply because it would have been impossible, due to the total separation of the groups by the abundance of a number of proteins. When total separation occurs, statistical analyses are useless. We could indeed have investigated one at a time all of the more than four thousand proteins, adjusting for age and sex, to verify whether other proteins were associated with the outcome. We instead concentrated on the identification of the stronger predicting proteins, because we believe the possibility of predicting the outcome by measuring a small number of proteins is a great achievement. We understand the curiosity of exploring further the behaviour of all other proteins, and we will leave this investigation to the next step of this study.
Insufficient Figure Captions (Figures 1 & 2): The captions for the Box plots (Figures 1 and 2) are overly simplistic. They lack essential information such as the definition of the box's horizontal lines (e.g., median), the definition of the whiskers, the statistical test used for comparison, and the meaning of any asterisks or p-values indicating statistical significance. This omission hinders the reader's ability to understand the data presented entirely.
Our reply: We added more information in the captions. However, as the Review can clearly see, the Box Plots do not report any p-value, and there is no asterisk to report any statistical significance. This is also due to the fact that, as clearly explained in the manuscript, the three groups are completely separated and, consequently, there is no need for a statistical test.
Insufficient Figure Captions (Figure 3): The caption for Figure 3 is insufficient. It requires a detailed explanation of what the size and color of the plots represent, as well as the meaning of the X-axis, to correctly interpret the bioinformatics enrichment analysis.
Our reply: We thank the reviewer for the observation regarding the clarity of Figure 3. In direct response, we have completely revised and replaced the original figure with an updated enrichment bubble plot that integrates additional significantly enriched GO terms across Biological Process (BP), Molecular Function (MF), and Cellular Component (CC) categories. The x-axis indicates the number of dysregulated proteins associated with each GO term (i.e., term count).
Insufficient Figure Captions (Figures 4 & 5): Similarly, Figures 4 and 5 need comprehensive explanations for the color and size of nodes and edges, and any other visual elements, to allow for proper interpretation of the Reactome pathways and Ingenuity Pathway Analysis network.
Our reply: We added more information in the captions. However, as the Review can clearly see, the Box Plots do not report any p-value, and there is no asterisk to report any statistical significance. This is also due to the fact that, as clearly explained in the manuscript, the three groups are completely separated and, consequently, there is no need for a statistical test.
Clarity in Three-Group Comparisons: When performing three-group comparisons, it must be clearly specified which groups are being compared (e.g., CTRL vs. low-grade EC, low-grade EC vs. high-grade EC, CTRL vs. high-grade EC). The methods for statistical comparison for each pair, including the specific tests and p-values, should be explicitly stated.
Our reply: As clearly stated in the manuscript, we did not use any statistical method to compare the three groups, simply because, when groups are completely separated, no statistical analysis can be performed. We understand that this is uncommon and can generate some surprise. However, this is the ideal case, in which we are beyond the usefulness of statistical methods. As clearly stated, 42 proteins completely separate controls from all EC cases taken together. In addition, 151 proteins completely separate high-grade from low-grade ECs. Finally, two proteins are able to completely separate the three groups.

Round 2
Reviewer 1 Report
Comments and Suggestions for Authors
The revised manuscript is significantly improved. It is appropriate for publication.
It would be preferable to explain in the results section the differences between the two figures showing Volcano plots.
The following statement should be removed
"as Spectronaut 19 does not support missing value imputation" Page 27 , lane 508-509
Spectronaut 19 allows missing value imputation, please see manual pages 114-116, 137
In my opinion missing value imputation for proteins that are identified in 40% of the samples gives reliable data and greatly improves the biological insights that can be obtained from a proteomics study. The authors do not want to use this approach, this is not optimum but acceptable. Other scientists will be able to use and reanalyse the data after the manuscript is published.
Author Response
Reviewer 1
It would be preferable to explain in the results section the differences between the two figures showing Volcano plots.
Our reply: We have clarified in the Results section the differences between the two figures showing the Volcano plots.
The following statement should be removed "as Spectronaut 19 does not support missing value imputation" Page 27 , lane 508-509.
Our reply: The indicated statement has been removed as requested.